# The Efficacy of Vitamin D Supplementation in the Treatment of Fibromyalgia Syndrome and Chronic Musculoskeletal Pain

**DOI:** 10.3390/nu14153010

**Published:** 2022-07-22

**Authors:** Mauro Lombardo, Alessandra Feraco, Morena Ottaviani, Gianluca Rizzo, Elisabetta Camajani, Massimiliano Caprio, Andrea Armani

**Affiliations:** 1Department of Human Sciences and Promotion of the Quality of Life, San Raffaele Open University, 00166 Rome, Italy; alessandra.feraco@uniroma5.it (A.F.); morena.ottaviani@uniroma5.it (M.O.); elisabetta.camajani@uniroma1.it (E.C.); massimiliano.caprio@uniroma5.it (M.C.); andrea.armani@uniroma5.it (A.A.); 2Laboratory of Cardiovascular Endocrinology, San Raffaele Research Institute, IRCCS San Raffaele Roma, Via di Val Cannuta, 247, 00166 Rome, Italy; 3Independent Researcher, Via Venezuela 66, 98121 Messina, Italy; gianlucarizzo@email.it; 4PhD Program in Endocrinological Sciences, Department of Experimental Medicine, University of Rome “La Sapienza”, Piazzale Aldo Moro, 5, 00185 Rome, Italy

**Keywords:** vitamin D, fibromyalgia, widespread pain, hypovitaminosis, systematic review

## Abstract

Fibromyalgia syndrome (FMS) and chronic widespread musculoskeletal pain (CMP) are diffuse suffering syndromes that interfere with normal activities. Controversy exists over the role of vitamin D in the treatment of these diseases. We carried out a systematic literature review of randomized controlled trials (RCT) to establish whether vitamin D (25OHD) deficiency is more prevalent in CMP patients and to assess the effects of vitamin D supplementation in pain management in these individuals. We searched PubMed, Physiotherapy Evidence Database (PEDro), and the Cochrane Central Register of Controlled Trials (CENTRAL) for RCTs published in English from 1 January 1990 to 10 July 2022. A total of 434 studies were accessed, of which 14 satisfied the eligibility criteria. In our review three studies, of which two had the best-quality evidence, a correlation between diffuse muscle pain and 25OHD deficiency was confirmed. Six studies, of which four had the best-quality evidence, demonstrated that appropriate supplementation may have beneficial effects in patients with established blood 25OHD deficiency. Eight studies, of which six had the best-quality evidence, demonstrated that 25OHD supplementation results in pain reduction. Our results suggest a possible role of vitamin D supplementation in alleviating the pain associated with FMS and CMP, especially in vitamin D-deficient individuals.

## 1. Introduction

Fibromyalgia syndrome (FMS) is a condition characterized by diffuse chronic musculoskeletal pain (CMP) associated with fatigue, sleep, memory, and mood problems. FMS is defined as muscle pain significantly reducing the quality of life of the affected person and interfering with normal daily activities. In CMP, organic damage is infrequently observed, whereas it is often reported in myofascial pain syndrome (MFPS) [1,2]. The prevalence of FMS in the general population is estimated to be 1.7% using the 1990 American College of Rheumatology (ACR) criteria and 5.4% using the 2010 modified version of the ACR criteria [3].

FMS can be defined as a form of extra-articular or soft-tissue “rheumatism” [4]. The pain manifests throughout the body, with different intensity levels, initially appearing in one area, such as the cervical spine or shoulders, and then spreading over time to other areas. Algic symptoms are reported to be constant and described in a variety of ways including burning sensation, paresthesia or dysesthesia, stiffness, tension, contracture, and heaviness of the limbs or spine, at rest and in motion. The pain can vary according to the time of day, activity levels, weather conditions, and sleep patterns; therefore, the dimensions of both physical and mental fatigue are variable [5]. In addition to pain, which is mainly felt in skeletal muscles, symptoms of general malaise are also reported; about 90% of patients with FMS report asthenia, along with marked “flu-like” exhaustion, reduced resistance to fatigue, headaches that may sometimes have a muscular tensive origin, or unspecified migraine. Patients with FMS very often experience sleep alterations, disturbances in the sleep–wake rhythm, sleep disturbances, or REM-phase disturbances [6]. However, it has recently been suggested that sleep disturbances and CMP may be included among the causative factors of FMS rather than its manifestation [7]. Thus, insomnia may precede the onset of pain and may be a predictor of its very persistence.

Moreover, myalgic encephalomyelitis/chronic fatigue syndrome (ME/CFS) is a complex and controversial clinical condition that causes cognitive impairment, myalgia, arthralgia, orthostatic intolerance, inflammation, and innate immunity disturbances. As reported by Rasa et al. [8], chronic viral infection as a cause of ME/CFS has long been debated. During the recent COVID-19 pandemic, cases of long or post-acute sequelae of COVID-19 (PASC), a series of chronic symptoms experienced by patients after the resolution of acute COVID-19, were detected, presenting a symptomatology similar to ME/CFS [9]. FMS, CMP, and ME/CFS have a negative impact on the quality of life of affected patients [10]. An important aspect of these diseases is the close correlation between stress and disease symptoms. Furthermore, the prevalence of psychiatric comorbidities, such as depression, anxiety disorders, or mood disorders, among patients with FMS reaches 60% [11]. Thus, it is still unclear whether psychological and psychic problems are responsible for disease development, thus triggering and increasing its symptoms, or whether unsolved pain leads to physical and mental stress with a reduction in the quality of life. This phenomenon drags the affected person into a situation of emotional and psychological instability. There are currently no instrumental or laboratory tests that can clearly diagnose FMS and CMP. However, CMP may be diagnosed through physical examination by an experienced practitioner. Since the symptoms of these illnesses are nonspecific and similar to those of other diseases, many patients undergo complicated and repetitive evaluations before a true diagnosis is made. It is, therefore, possible that FMS may be linked to dysfunctional pain processing in the brain. In support of this, FMS and ME/CFS patients become hypersensitive to pain and develop “hypervigilance”, which may also be associated with autoimmune and inflammatory markers, as well as neurotransmitter changes [12].

The etiology of FMS and CMP remains unknown, and, despite the various pathogenetic hypotheses, a multifactorial pathogenesis based on a genetic predisposition [2,13,14] has been proposed. As reported by Sluka and Clauw [15], FMP and CMP have significant alterations in central nervous system factors that lead to increased pain and sensory processing. There may also be alterations in the immune system that lead to an increased inflammatory state, and there are a number of other behaviors such as sleep dysfunction, mood difficulties, and fatigue that contribute to pain and dysfunction in people with fibromyalgia. In recent years, imbalances in chemical mediators such as neurotransmitters at a central level and endocrine axis dysfunction have been demonstrated [16], while another study observed muscle fragility following repeated microtrauma [13]. Patients with FMS produce higher levels of free radicals compared to the healthy population and have a decreased endogenous antioxidant ability with a consequent increase in inflammation [2]. The observed central sensitization could be partially explained by increased ROS-susceptible lipid abundance in the central nervous system [14]. The development and progression of FMS could, therefore, also depend on increased pro-oxidant substances in both nerves and other soft tissues [15]. A large body of evidence indicates that vitamin D (25OHD) deficiency may be related to an enhanced risk of FMS and CMP [17,18]. Accordingly, clinical studies showed that individuals with FMS display lower 25OHD circulating levels compared to controls [19,20,21]. The vitamin D receptor is widely expressed in muscles, providing a direct regulatory role in this tissue. 25OHD is involved in myogenesis, cell proliferation, differentiation, protein synthesis, mitochondrial metabolism regulation, and endothelial function [22,23]. Moreover, 25OHD deficiency is probably involved in the etiology of chronic pain [24] because this condition induces muscle impairment in terms of both tissue structure and function. These disorders have been defined as “vitamin D deficiency myopathies” and consist of muscle atrophy or sarcopenia, problems in coordination and balance, with increased risk of falls, and chronic and widespread muscle pain [25,26]. However, the effectiveness of vitamin D supplementation in CMP patients is currently still debated.

The aim of this review is to summarize the literature investigating the effectiveness of vitamin D supplementation in CMP and FMS symptoms.

## 2. Materials and Methods

### 2.1. Research Strategy

This systematic review was performed in accordance with the Preferred Reporting Items for Systematic Reviews and Meta-Analyses (PRISMA) guidelines [27]. PubMed, Physiotherapy Evidence Database (PEDro), and the Cochrane Central Register of Controlled Trials (CENTRAL) were searched. The following keywords were used: “vitamin D deficiency” OR “vitamin D” OR “vitamins” OR “25OHD” OR “cholecalciferol” OR “vitamin intake” OR “vitamin supplementation” OR “supplementation” OR “integration” OR “vitamin D replacement therapy” AND “fibromyalgia” OR “musculoskeletal pain” OR “chronic widespread pain”. We researched papers from 1 January 1990 to 10 July 2022.

### 2.2. Inclusion and Exclusion Criteria

After conducting the literature search in the identified databases, an initial screening of the three result lists was carried out by reading the title and eliminating meta-analyses, systematic reviews, cross-sectional studies, abstracts, case reports, case series, conference proceedings, reviews, letters, short surveys, books, and book chapters. Only RCT studies were taken into consideration for this review. Duplicate studies identified in multiple databases were then eliminated. Studies on functional foods, supplements, or probiotics, or that were performed in animal models or children were also deleted. The remaining articles were subjected to additional examination based on abstract and type of study, in order to establish the eligibility of each paper for the assessment. Only clinical studies were included in the systematic review. All documents were combined into one file, and redundant records were removed after being manually checked. Studies without control groups, with pain scale assessment after zoledronic acid injection or vitamin D supplementation, including acupuncture treatment according to the principles of traditional Chinese medicine, or using a combination of different nutraceuticals were excluded. The article selection process is depicted in the flowchart in the results (Figure 1). Two reviewers (M.L. and M.O.) evaluated the adequacy of inclusion criteria separately. Where there was disagreement, a third reviewer (A.A.) was included in the review procedure. In case of missing data, the author of the study was contacted, or any supplementary files published with the article were consulted.

### 2.3. Quality Assessment

The quality of the eligible studies was assessed using the Cochrane risk-of-bias tool (version 22 August 2019) [28]. The tool includes five domains covering all types of biases that may influence the results of randomized trials. The risk of bias for each study was scored in accordance with five domains: D1 = errors due to the randomization process; D2 = errors due to the administration of the interventions; D3 = errors due to the reporting of missing data; D4 = errors due to outcome assessment; D5 = errors due to selective reporting of results. If the study was considered to be at low risk of bias for all domains for the outcome, it was classified as “low risk of bias”. In cases where the study was deemed to raise some concerns in at least one domain for the outcome, but was not at high risk of bias for any domain, it was rated as “some concerns”. “High risk of bias” was attributed to studies with an elevated risk of bias in at least one domain for the outcome.

## 3. Results

### 3.1. Eligibility

A total of 434 studies were extracted from the database and manual search, 35 of which were excluded because they were duplicates. Of the 399 remaining papers, 372 were discarded because they were reviews, abstracts, or nonclinical studies. The full texts of the 27 remaining studies were read, and their content was cross-referenced with the inclusion criteria. A total of 14 studies met the eligible criteria [29,30,31,32,33,34,35,36,37,38,39,40,41,42]. Details regarding the included studies are summarized in Table 1.

### 3.2. VAS Scale

Nine RCT studies (Table 2) assessed changes in patients’ pain symptoms induced by vitamin D supplementation using the visual analogue scale (VAS). VAS is a pain assessment scale first created by Hayes and Patterson in 1921 [43]. The ratings are based on self-reported pain scores that are recorded with a single handwritten mark set at a point on a straight horizontal line of fixed length, usually 100 mm, which provides a line of continuum across the two ends of the scale: ‘no pain’ at the left (0 mm) end of the range and ‘worst pain’ at the right (100 mm) end of the scale.

Knutsen et al. [32] showed that there was no significant effect of vitamin D supplementation over placebo on pain parameters, as proven by equal pain levels in the last 2 weeks, number of pain sites, total VAS score, headaches in the last 2 and 4 weeks, HIT-6 score, or VAS. There was no significant difference between vitamin D and placebo groups in subjects with chronic lower-back pain (CLBP) who were treated with home-exercise and celecoxib prescriptions [34]. Similarly, Lozano-Plata [42] did not observe improvements in VAS or Fibromyalgia Impact Questionnaire (FIQ) in vitamin D-supplemented subjects compared with placebo. Furthermore, in the study by Scheuder et al. [31], VAS scores for pain did not improve significantly after vitamin D supplementation, but the authors found a small positive effect on the Likert pain self-assessment scale. On the other hand, different studies demonstrated the effectiveness of vitamin D in the treatment of FMS and CMP, with a marked reduction in pain, as assessed by VAS, during the treatment period. Indeed, optimization of calcifediol levels had a favorable influence on pain perception [33]. Accordingly, the study by Gendelman et al. [35] showed that the group receiving 4000 daily units of vitamin D had a statistically greater decline in their VAS score for pain than the placebo group. Similarly, nonspecific CMP, functional capacity, and quality of life in obese women were improved by a protocol involving vitamin D supplementation, combined with aerobic exercise in addition to a well-balanced low-calorie diet [40]. Sakalli et al. demonstrated that 300,000 IU of vitamin D per os or parenteral administration decreased pain in the elderly [30]. Interestingly, in this study, parathormone levels, but not vitamin D concentrations, showed significant correlations with reduced VAS.

### 3.3. Other Pain Scales

In addition to VAS, other pain assessment scores were used to investigate the potential impact of vitamin D supplementation on CMP and FMS. These studies are shown in Table 3.

The study by Wu et al., which lasted for 3.3 years, demonstrated that a monthly dose of 100,000 IU vitamin D did not significantly improve pain, nor the odds of prescribing analgesics [37]. In patients with 25OHD deficiency at baseline, vitamin D supplementation led to fewer nonsteroidal anti-inflammatory drug (NSAID) prescriptions compared to placebo group, but similar benefits were not observed for other analgesic outcomes. High-dose supplementation with vitamin D for 6 weeks had a small positive effect compared with the placebo group (34.9% vs. 19.5%, *p* = 0.04) on persistent nonspecific CMP in non-Western immigrants in the Netherlands [31]. The study by Aldaoseri et al. [38] showed that 25OHD deficiency is common in FMS patients and is also associated with exacerbation of FMS symptoms. In this study, vitamin D supplementation for 12 weeks was associated with significantly improved outcomes in 25OHD-deficient patients. Another study demonstrated that the initial level of 25OHD, as well as monthly dosing of vitamin D treatment, could be crucial to reduce chronic pain. Indeed, a significant effect on the McGill Pain Map in elderly patients with vitamin D deficiency treated with a monthly dose of 24,000 IU vitamin D was observed [39]. Sakalli et al. demonstrated that a single megadose of vitamin D (300,000 IU) was effective in decreasing CMP, measured by the bodily pain test (BP), in the elderly [30]. Two studies evaluated the efficacy of combining vitamin D and other tools to reduce symptoms in patients suffering from FMS and CMP. A treatment with a vitamin D supplement and a standard antidepressant prescribed to FMS patients with 25OHD deficiency could lead to significant benefits in terms of both physical and psychological symptoms [36]. The combined benefit of vitamin D supplementation and physiotherapy was significantly higher than physiotherapy alone in improving pain-related symptoms (−4.92 vs. −3.79; *p* < 0.001) in CMP patients [41].

### 3.4. Bias

The risk of bias in the primary studies based on the Cochrane risk-of-bias tool is shown in Figure 2. There was “high risk” of bias in six studies [29,32,38,39,41,42], whereas six papers had “low risk” [33,34,35,36,37,40], and two manuscripts had “some concerns” [30,31]. The overall risk of bias is illustrated in Figure 3. The most frequent source of a high of risk bias was D5, i.e., the “selection of the reported result”, which was present in three studies out of 13, i.e., in about 23% of the cases. This was followed by D1, D2 and D3, each present in two studies with a percentage of around 15%. D4 was at the back of the list, appearing with high risk in only one study.

## 4. Discussion

The therapeutic approach to FMS and CMP necessarily involves a multidisciplinary protocol, which should be conducted as a team because of the many facets of the pathologies and the many compromised systems [44,45]. The severity of the pain and the lack of treatment protocols often lead the patient to self-medicate [46]. Gabapentinoids (gabapentin and pregabalin) [47], antidepressants (tricyclic antidepressants and serotonin noradrenaline reuptake inhibitors) [48], and melatonin [49] are often used for obtaining a direct myorelaxant effect and inducing a state of psychic relaxation, which has repercussions on the state of muscular tension, thus reducing pain. Indeed, a similar improvement occurs even when different treatments have been proposed, such as pain neuroscience education and physical training [50], acquisition of knowledge of pain neurophysiology [51], placebo [37,41], or alternative treatments [34,38]. The European League Against Rheumatism (EULAR) emphasizes the importance of a nonpharmacological approach to the treatment of FMS, e.g., through nutritional interventions, including vitamin D supplementation [52].

In our review, three studies [31,35,38], of which two had the best-quality evidence, confirmed a correlation between diffuse muscle pain and 25OHD deficiency. An increased incidence of chronic pain in subjects with low 25OHD levels was previously demonstrated in studies performed in the early 2000s [53,54]. In another manuscript, this association was only confirmed in women, although the authors did not exclude that different ways of managing the pain between men and women with CMP could explain these results [55]. Heidari et al. reported various types of CMP in patients with 25OHD deficiency, particularly in women [56]. Other studies showed that the association between reduced levels of vitamin D and chronic pain is not restricted only to women. Knutsen et al. correlated hypovitaminosis D with chronic widespread pain in a multiethnic population [57]. Another study [58] conducted in men demonstrated that patients with FMS have a significantly lower serum 25OHD concentration on average than healthy controls (23.9 ng/mL vs. 25.6 ng/mL; *p* = 0.05). This association remained valid after adjusting for physical activity, smoking, and alcohol use, thus confirming the relationship between low levels of 25OHD and pain, regardless of other factors [58].

Deficiency of 25OHD may play a role in the etiology of chronic pain conditions such as headaches or abdominal pain [59,60], and a significant correlation was indeed revealed between low 25OHD levels and increased opioid consumption [61].

The correlation between hypovitaminosis D and CMP has been explained by the phenomenon of central sensitization of pain processing, with non-physiological responses to even mild pain stimuli [62]. The role of vitamin D in modulating pain signaling pathways can be explained by the expression of vitamin D receptor and/or the enzymes regulating vitamin D activity in tissues involved in tissues participating in pain sensing and processing, such as the skin, dorsal root ganglia, and spinal cord [62]. It has also been hypothesized that 25OHD may have anti-inflammatory properties that contribute to pain relief. In vitro studies have shown that vitamin D can reduce the synthesis of prostaglandin E2 (PGE2) to under-regulate proinflammatory pathways [63]. More in detail, vitamin D treatment of epithelial cells was found to reduce the expression of IL-1β-induced microsomal PGE synthase and stimulate the PGE2-degrading enzyme 15-hydroxy PG dehydrogenase [63]. Furthermore, 1,25-(OH)_2_D_3_ has been shown to exert anti-inflammatory effects on macrophages, increasing interleukin (IL)-10 and reducing inflammatory cytokine production. A more tolerogenic phenotype is also induced in dendritic cells by vitamin D [64] (Figure 4).

These data suggest that hypovitaminosis D activates inflammatory cytokines that may modulate central and peripheral pain perception [64]. Alternatively, a different process may be suggested; patients with CMP may be 25OHD deficient due to their lifestyle, poor mobility resulting from the pain experienced, or associated depression that would potentially lead patients to spend less time outdoors, with a tendency to accumulate fat mass. In overweight subjects, 25OHD is often reduced because the high levels of adiposity sequester vitamin D and make it unavailable [65]. In accordance, in a previous study, we demonstrated in obese patients that weight loss leads to an increase in circulating 25OHD even without supplementation [66]. The present systematic literature review identified three studies [29,32,42] that observed scarce analgesic effects of vitamin D supplementation which did not improve VAS scores. Nevertheless, data from these studies showed a high risk of bias.

Our analyses led to the identification of six studies, of which four had the best-quality evidence [30,31,34,36], demonstrating that appropriate supplementation may have beneficial effects for CMP in patients with established blood 25OHD deficiency. The 25OHD threshold level and ideal dosage are nevertheless still under study and need further investigation. Controversial is still the optimal route of administration (oral or intramuscular) of vitamin D, with data by Gupta et al. suggesting that intramuscular replacement of 25OHD might result in a more prolonged increase from baseline levels [67]. Although almost all studies have proposed a dosage of 50,000 IU/week per os, the correct mode of administration, the appropriate amount, and the frequency and the duration of treatment in the case of CMP are not yet clear. In all considered studies, a marked improvement in 25OHD levels was found after supplementation either through daily oral supplementation or in monthly megadoses, both orally and intramuscularly. Notably, a monthly high-dose vitamin D supplementation does not prevent falls or fractures, whereas an annual high-dose vitamin D in elderly women even may result in an increased risk of falls and fractures, suggesting potential adverse outcome of high-dose vitamin D supplementation [68,69].

The heterogeneity of 25OHD threshold values was also highlighted in our review. Indeed, the threshold value for 25OHD deficiency ranges from 10 to 32 (ng/mL) (Table 1). The main cutoffs proposed in the last 10 years by the guidelines consider a cutoff of 10 ng/mL for severe vitamin D deficiency, as 25OHD levels below 10 ng/mL in the presence of a diet with adequate calcium intake have been found to be associated with an increased prevalence of rickets and osteoporosis [70,71,72]. Several studies [73,74] showed that 25OHD levels below 20 ng/mL may be correlated with secondary hyperparathyroidism, and that PTH levels reached a plateau for 25OHD levels between 30 and 40 ng/mL. In contrast, other authors have not demonstrated a “J” curve for the relationship between 25OHD and PTH, suggesting that numerous other factors such as age, gender, degree of obesity, and ethnicity come into play in the steady state of 25OHD. Interestingly, vitamin D treatment of subjects with serum levels of 25OHD higher than 50 nmol/L does not reduce the risk of cancer, cardiovascular events, hypertension, and type 2 diabetes, excluding healthy benefits of vitamin supplementation in vitamin D-replete individuals. These data are not in contrast with the recommendation of vitamin D supplements in subjects with suboptimal serum levels of 25OHD, of course. Intriguingly, apart from the context of FMS and CMP, a number of studies have explored the potential effects of vitamin D on physical performance in healthy subjects, showing that vitamin D levels are associated with muscle performance metrics [75,76]. Sarcopenia and impaired muscle strength have also been related to vitamin D deficiency [77,78]. On the other hand, vitamin D supplementation has been associated with improved aerobic and anaerobic capacity of healthy subjects [79], even though other studies did not observe improvements in muscular performance [80,81], suggesting that the impact of vitamin D supplements on exercise or sport performance has yet to be clarified.

## 5. Conclusions

In conclusion, this review suggests that vitamin D deficiency is frequently observed in FMS and CMP patients, and supplementation with vitamin D can be proposed to reduce musculoskeletal pain and improve the quality of life in vitamin D-deficient subjects with FMS and CMP.

## Figures and Tables

**Figure 1 nutrients-14-03010-f001:**
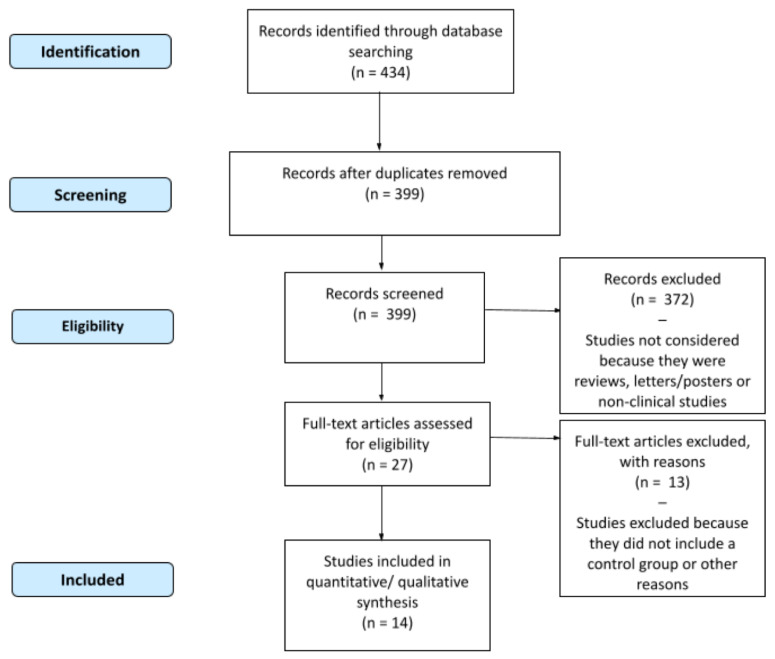
Flowchart (PRISMA) of studies included.

**Figure 2 nutrients-14-03010-f002:**
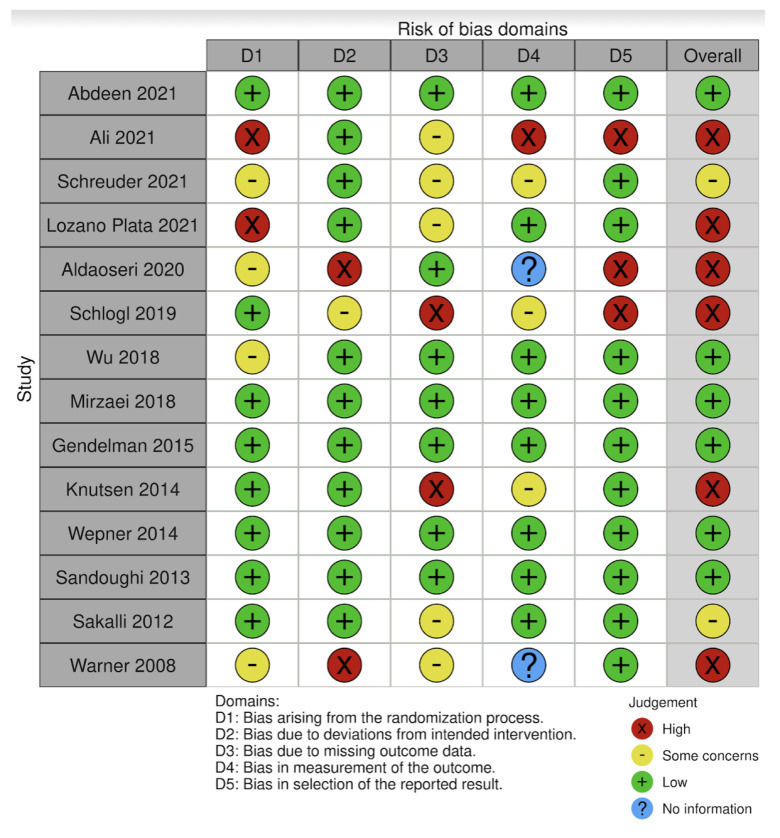
Algorithm adopted in the process of defining the judgement to be assigned to each domain. The possible judgments of risk of bias were ‘low risk of bias’, ‘some concern’, or ‘high risk of bias’. The acronyms for the answers to the questions had the following explanation: Y/PY = “yes” or “probably yes”; N/PN = “no” or “probably no”; NI = “no information” [29,30,31,32,33,34,35,36,37,38,39,40,41,42].

**Figure 3 nutrients-14-03010-f003:**
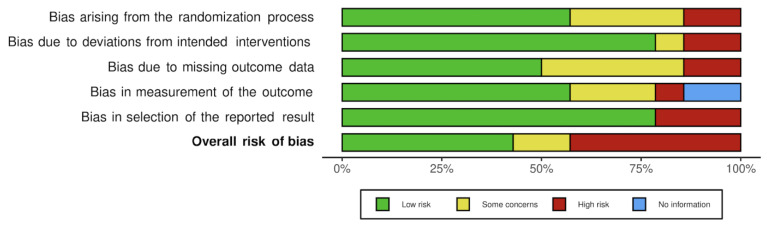
Risk-of bias-assessment using the Cochrane Collaboration tool.

**Figure 4 nutrients-14-03010-f004:**
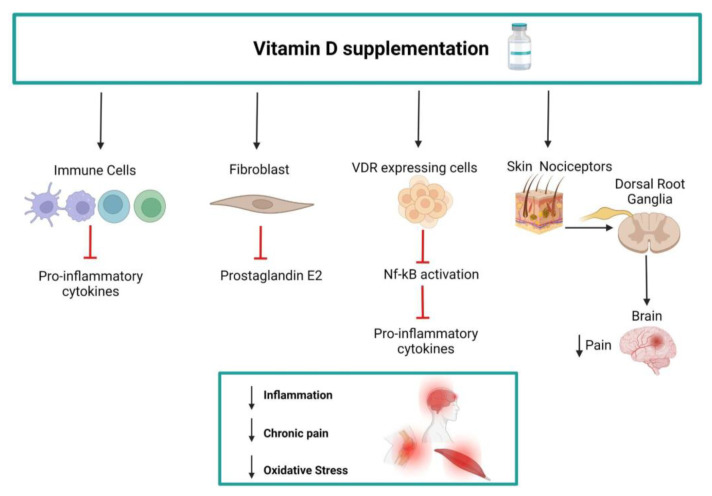
Possible mechanisms underlying the efficacy of vitamin D supplementation in chronic pain management include anti-inflammatory effects mediated by reduced cytokine and prostaglandin release and effects on immune cell responses. Furthermore, alterations in the somatosensory nervous system are observed in fibromyalgia. Pain signals are transmitted to the brain by pain receptors called nociceptors distributed also in the skin. Vitamin D and vitamin D receptor (VDR) activation has been described in several tissues, including skin, DRG neurons, and the brain, where the pain signal is perceived, thus contributing to ascending pathways that mediate chronic pain.

**Table 1 nutrients-14-03010-t001:** Main features of included studies.

										Sample Size		Age Mean		Age Mean	Vitamin D Level (ng/mL)-Baseline		Vitamin D Level (ng/mL)-End of Treatment				
Ref.	First Author	Year	Country	Study Design	Topic	Gender	Season of Measurement	Duration (Weeks)	Control Characteristics	Intervention Group	Control	Intervention Group	Control	*p*-Value	Intervention Group	Control	Intervention Group	Control	Dosage (Weekly IU)	Threshold of Vit D Deficiency (ng/mL)	Method of Vit D Measure
[29]	Warner	2008	USA	RCT	CMP	F	?	12	Diffuse pain/low 25OH Vit D	20	22	58.0 ± 7.3	56.7 ± 11.3	0.634	16.8 ± 2.9	15.9 ± 3.6	?	?	50,000	<20	?
[30]	Sakalli	2012	Turkey	RCT	CMP	M + F	Winter–spring	4	Placebo	30	30	69.8 ± 3.7	68.9 ± 2.7	ns	20.9 ± 9.5	21.2 ± 7.4	27.0 ± 12.0	21.0 ± 5.5	300,000	<12.5	RIA
[31]	Scheuder	2012	Netherlands	RCT ç	CMP	M + F	?	12	Placebo	44	40	42.9 ± 9.5	40.8 ± 11.3	ns	19.7 ± 10.7	19.7 ± 8.8	20.2 ± 10.3	19.6 ± 9.0	25,000	<20	RIA
[32]	Knutsen	2014	Norway	RCT °	CMP	M + F	Winter–spring	16	Placebo	84/85	82	37.5 ± 8.1	38.0 ± 7.2	?	28.7 ^	29.2 ^	48.8 ^	27.5 ^	(1) 7000 (2) 2800 $	?	LC/MS/MS
[33]	Wepner	2014	Austria	RCT	FMS	M + F	Summer	49	Placebo	15	15	49.1 ± 5.7	47.6 ± 4.9	0.438	19.0 ± 5.9	20.9 ± 6.3	26.3 ± 6.9	26.3 ± 11.7	#	<32	?
[34]	Sandoughi	2015	Iran	RCT	CMP	M + F	Spring–autumn	8	Placebo	26	27	33.2 ± 6.5	33.2 ± 6.5	0.430	17.9 ± 9.0	19.8 ± 9.6	27.5 ± 9.0	18.9 ± 7.8	50,000	<20	ELISA
[35]	Gendelman	2015	Israel	RCT	CMP	M + F	Autumn–spring	12	Placebo	38	36	56.8 ± 13.1	57.3 ± 13.8	0.885	21.8 ± 8.7	25.5 ± 13.0	31.7 ± 22.8	24.0 ± 13.1	28,000	<30	RIA
[36]	Mirzaei	2018	Iran	RCT	FMS	M + F	?	8	Antidepressant	37	37	42.1 ± 10.8	41 ± 10.3	?	11.4 ± 6.5	13.4 ± 7.3	33.5 ± 12.2	13.3 ± 7.2	50,000	<20	RIA
[37]	Wu	2018	New Zealand	RCT §	CMP	M + F	?	3.3 yrs	Placebo	2558	2550	65.9 ± 8.3	65.9 ± 8.3	ns	26.6 ± 9.0	26.3 ± 9.0	54.1 ± 16.0	26.4 ± 11.6	25,000	<20	LC/MS/MS
[38]	ALdaoseri	2019	Iraq	RCT	FMS	M + F	Spring–autumn	12	Antidepressant	53	53/54	34.3 ± 9.5	35.1 ± 11.6	?	?	?	?	?	50,000	<20	HA
[39]	Schlögl	2019	Switzerland	RCT °	CMP	M + F	Year	52	Vitamin D 800 IU/day	66	67/67	77 ± 4.7	78 ± 5.3	?	18.4 ± 7.6	20.9 ± 9.2	?	?	(1) 14,000 (2) 2800 + 75 mg of calcifediol (3) 5600 &	<20	LC/MS/MS
[40]	Abdeen	2021	Egypt	RCT °	CMP	F	Winter–spring	12	Aerobic training	15	15/15	34.8 ± 2.6	35.4 ± 2.7	0.830	17.1 ± 1.2	16.4 ± 1.7	23.9 ± 4.3	17.6 ± 2.3	50,000	<10	BioPlex
[41]	Ali	2021	Bangladesh	RCT	CMP	M + F	Year	8	Physiotherapy	72	63	51.2 ± 13.3		0.065	?	?	?	?	50.000	<20	CMIA
[42]	Lozano-Plata	2021	Mexico	RCT	FMS	F	?	12	Placebo	40	40	50.3 ± 11.9	51.4 ± 9.5	ns	20.1 ± 14.5	12.6 ± 13.4	51.1	20.8	50.000	<20	ELFA

Ç, semi-crossover study; §, post hoc analysis; ^, combined intervention group; ° 3 equally sized groups; ?, information not available. M, male; F, female; RCT, randomized controlled studies; $, daily administration; #, depending on their serum calcifediol levels, the verum group received 2400 IU (serum calcifediol levels <60 nmol/L) or 1200 IU (serum calcifediol levels 60 to 80 nmol/L) of cholecalciferol (vitamin D3) daily; &, Three study groups with monthly treatments: a low-dose control group of vitamin D (24,000 IU vitamin D3/month), a high dose of vitamin D3 (60,000 IU vitamin D3/month), or a combination of calcifediol and vitamin D3 (24,000 IU vitamin D3 plus 300 mg calcifediol/month). Abbreviations: FMS, fibromyalgia; CMP, chronic musculoskeletal pain; VAS, visual analog scale; FPS, functional pain score; RIA, radioimmunoassay; ELISA, enzyme-linked immunosorbent assay; LC/MS/MS, liquid chromatography–tandem mass spectrometry; HA, hormone analyzer; BioPlex, BioPlex^®^ 2200 Fully Automated Immunoassay Instrument 2200 System; CMIA, chemiluminescence microparticle immunoassay; ELFA, enzyme-linked fluorescent assay.

**Table 2 nutrients-14-03010-t002:** Studies that evaluated the effect of vitamin D in patients with CMP or FMS applying the VAS scale.

		VAS—Baseline		VAS—End of Treatment			Association between Low Vitamin D and MS Pain	Effects of Vitamin D Supplementation	Effects of Vitamin D Supplementation in People with Vitamin D Insufficiency at Baseline
Ref.	First Author	Case	Control	Case	Control	*p*-Value			
[29]	Warner	67.8 ± 22.0	67.3 ± 22.8	64.7 ± 18.0	53.6 ± 26.8	0.727	=	=	=
[30]	Sakalli	59 ± 24	59 ± 27	51 ± 23	55 ± 28	< 0.01		↑ *	
[31]	Scheuder	60.6 ± 21.2	65.1 ± 17.2	65.2 ± 16.1	66.0 ± 17.1	ns	↑ @	=	
[32]	Knutsen	22.3	25.2	14	14.3	0.18	=	=	
[33]	Wepner	62.0 ± 20.3	55.2 ± 20.5	68.7 ± 12.5	55.2 ± 21.8	0.025		↑	↑ §
[34]	Sandoughi	54.2 ± 16.5	64.4 ± 16.2	30.3 ± 31.4	31.1 ± 30.8	<0.001		=	
[35]	Gendelman	72.2 ± 20.6	63.3 ± 22.9	48.6 ± 26.0	54.6 ± 28.3	0.612	↑	↑	
[40]	Abdeen	61 ± 8.9	64 ± 8	27.5 ± 12.5	30.3 ± 12.5	0.001		↑ #	
[42]	Lozano-Plata	60 ± 30	60 ± 35	49	51.9	0.705		=	

Legend: =, no effects; ↑, positive effect; ns, not significant (*p* < 0.05). * Both in single intramuscular and oral megadose; @, small positive effect; § optimization of calcifediol levels in FMS had a positive effect on the perception of pain; # vitamin D supplementation combined with aerobic exercise in addition to a well-balanced hypocaloric diet. VAS (visual analog scale) indicates pain on visual analog scale (0–100 mm): 0 to 4 mm, no pain; 5 to 44 mm, mild pain; 45 to 74 mm, moderate pain; 75 to 100 mm, severe pain. FMS, fibromyalgia syndrome; CMP, diffuse chronic musculoskeletal pain.

**Table 3 nutrients-14-03010-t003:** Studies that used other pain scales than the VAS to assess the correlation between vitamin D deficiency and CMP pain and the efficacy of vitamin D supplementation in these patients.

Ref.	First Author	Pain Scale	Association between Low Vitamin D and CMP Pain	Effects of Vitamin D Supplementation	Effects of Vitamin D Supplementation in People with Vitamin D Insufficiency at Baseline
[29]	Warner	FPS			=
[30]	Sakalli	BP			↑ *
[31]	Scheuder	Likert scale	↑ @	↑ @	↑ @
[33]	Wepner	FIQ		=	
[35]	Gendelman	MPQ	↑	=	
[36]	Mirzaei	WPI, FIQ		↑ &	
[37]	Wu	PIQ-6		=	↑ $
[38]	ALdaoseri	WPI, SSS	↑	↑	↑
[39]	Schlögl	MPQ		=	↑
[41]	Ali	BPI		↑	
[42]	Lozano-Plata	FIQ		=	

Abbreviations: FMS, fibromyalgia syndrome; CMP, diffuse chronic musculoskeletal pain; FPS, functional pain score; BP, bodily pain scale (part of the SF-36 survey); FIQ, Fibromyalgia Impact Questionnaire; MPQ, McGill Pain Questionnaire; WPI, Widespread Pain Index; PIQ-6, Pain Impact Questionnaire; SSS, Symptom Severity Scale; BPI, Brief Pain Inventory scale. Legend: =, no effects; ↑, positive effect; ns, not significant (*p* < 0.05). * Both in single intramuscular and oral megadose; @, small positive effect; &, combination of trazodone and vitamin D; $, lower risk of nonsteroidal anti-inflammatory drug prescription.

## Data Availability

Data are available from the authors upon reasonable request.

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
