# Peer review of "The Efficacy of Vitamin D Supplementation in the Treatment of Fibromyalgia Syndrome and Chronic Musculoskeletal Pain"

_nutrients, 2022, doi:10.3390/nu14153010_

Round 1

Reviewer 1 Report

This is an important topic of study and the up to date best practices for pain, the mechanisms of vitamin D including some confounding issues around vitamin D insufficiency and supplementation. The assumptions made in the papers cited must be assessed for their validity given the progress of science. 

Author Response

This is an important topic of study and the up to date best practices for pain, the mechanisms of vitamin D including some confounding issues around vitamin D insufficiency and supplementation. The assumptions made in the papers cited must be assessed for their validity given the progress of science.

We thank the reviewer for the comment, which is undoubtedly appropriate. We considered the possibility that some of the studies included in our review might have BIAS. For this reason, we performed a bias analysis using the “Cochrane risk of bias tool”. In connection with your right observations on Warner's study, the BIAS analysis showed a high risk in this paper (figure 2). We have rewritten part of the discussion to point out that many studies have methodological flaws that affect their validity. 

Warner's study did not indicate an association between low levels of vitamin D with diffuse musculoskeletal pain. However, the score of this paper in term of high risk of bias (Figure 2) suggests that the provided data are less robust that those shown by Gendelman’ and Schreuder’ studies showing “best quality evidence” and indicating a “correlation between diffuse muscle pain and 25OHD deficiency”  As mentioned above, in the new version of the manuscript, we further pointed out that some studies have methodological flaws that affect their validity. 

Concerning the effects of supplementation with vitamin D, a high risk of bias was shown by 3 papers, including Warner's study, in the new version of the manuscript, and we mentioned such an issue in the section Discussion. We added to the text “The present systematic literature review has identified 3 studies that observed scarce analgesic effects of vitamin D supplementation which did not improve VAS scores. Nevertheless, data from these studies show a high risk of bias” We also deleted “The study by Warner et al. [24] showed that patients with CMP did not display a higher incidence of 25OHD insufficiency than controls. Thus, the same authors point out that normalisation of 25OHD levels through supplementation showed no effect on CMP”.

We also modified other parts of the text following the reviewer's comments in the manuscript.

Reviewer 2 Report

Thank you for a very clear and thorough process in presenting the need/appropriateness of the review using systematic review strategies.  It seems appropriate that additional information with regard to what level of 25OHD identified as low would be helpful as well as other specific details.

The 'tight' requirements for inclusion in the systematic review added strength to the results noted in the review.  It seems the a .034% inclusion rate from all of the available publication noted provided very specific information using RCT for the outcome statement for the project.

The abstract indicated two purposes (lines 4 of abstract)  or foci and the introduction focused just on one. (last line of introduction)  I mention this because, it would or could be beneficial to indicate responses to both in the conclusion. Perhaps a notation as to whether or not there was a difference noted through the review of studies with regard to prevalence of the deficit of vitamin D.

The discussion section of this was very thorough and clearly presented.  Do wonder if there is a connection in the role of this vitamin in the fitness focused population with regard to many of the points noted.

Author Response

Thank you for a very clear and thorough process in presenting the need/appropriateness of the review using systematic review strategies. 

We thank the reviewer for the positive comment on our work.

It seems appropriate that additional information with regard to what level of 25OHD identified as low would be helpful as well as other specific details.

We thank the reviewer for the precious comment. In the new version of the manuscript, we added a paragraph discussing the heterogeneity of threshold values for vitamin D deficiency.  

The 'tight' requirements for inclusion in the systematic review added strength to the results noted in the review.  It seems that a .034% inclusion rate from all of the available publications noted provided very specific information using RCT for the outcome statement for the project.

We thank the reviewer for this comment.

The abstract indicated two purposes (lines 4 of abstract)  or foci and the introduction focused just on one. (last line of introduction)  I mention this because, it would or could be beneficial to indicate responses to both in the conclusion. Perhaps a notation as to whether or not there was a difference noted through the review of studies with regard to prevalence of the deficit of vitamin D.

We thank the reviewer for constructive comments. We have revised the entire text to address the purposes written in the abstract and introduction.

The discussion section of this was very thorough and clearly presented.  Do wonder if there is a connection in the role of this vitamin in the fitness focused population with regard to many of the points noted.

As requested by the reviewer, we have added a paragraph in the discussion describing the role of vitamin D in the trained population.

Reviewer 3 Report

The manuscript entitled The Efficacy Of Vitamin D Supplementation In The Treatment Of Fibromyalgia Syndrome And Chronic Musculoskeletal Pain. The following is my comment and suggestions.

I feel that the manuscript requires a revision and must be given to improve the quality of Figures (especially ambiguous-looking of Figure 2). Moreover, a detailed description of the role and therapeutic effects of vitamin D on fibromyalgia syndrome and chronic musculoskeletal pain with a new Figure should be well presented in this manuscript. 

Author Response

The manuscript entitled “The Efficacy Of Vitamin D Supplementation In The Treatment Of Fibromyalgia Syndrome And Chronic Musculoskeletal Pain”. The following is my comment and suggestions.

I feel that the manuscript requires a revision and must be given to improve the quality of Figures (especially ambiguous-looking of Figure 2).

We thank the reviewer for the comment, which is undoubtedly true. We have removed figure 2. It was the diagram present in the bias evaluation guide and did not offer any interesting insights to the paper but only provided more confusion. Thanks again for the helpful suggestion. 

Moreover, a detailed description of the role and therapeutic effects of vitamin D on fibromyalgia syndrome and chronic musculoskeletal pain with a new Figure should be well presented in this manuscript. 

As suggested by the Reviewer, we added a new figure (figure 4)  in the revised version of the manuscript,  to describe the potential mechanisms underlying the efficacy of Vitamin D supplementation in managing  fibromyalgia syndrome and chronic musculoskeletal pain. 

Round 2

Reviewer 3 Report

All the revisions have been completed.